# Applying Grammar-based Compression to RDF

Michael Röder[1][2][0000−0002−8609−8277], Philip Frerk[1], Felix Conrads[1], and
Axel-Cyrille Ngonga Ngomo[1][2][0000−0001−7112−3516]

[1] DICE group, Department of Computer Science, Paderborn University
`michael.roeder|axel.ngonga@upb.de`
[2] Institute for Applied Informatics, Leipzig, Germany

**Abstract.** Data compression for RDF knowledge graphs is used in an increasing number of settings. In parallel to this, several grammar-based graph compression algorithms have been developed to reduce the size of graphs. We port gRePair—a state-of-the-art grammar-based graph compression algorithm—to RDF (named RDFRePair). We compare this promising technique with respect to the compression ratio to the state-of-the-art approaches for RDF compression dubbed HDT, HDT++ and OFR as well as a $k^2$-trees-based RDF compression. We run an extensive evaluation on 40 datasets. Our results suggest that RDFRePair achieves significantly better compression ratios and runtimes than gRePair. However, it is outperformed by $k^2$ trees, which achieve the overall best compression ratio on real-world datasets. This better performance comes at the cost of time, as $k^2$ trees are clearly outperformed by OFR w.r.t. compression and decompression time. A pairwise Wilcoxon Signed Rank Test suggests that while OFR is significantly more time-efficient than HDT and $k^2$ trees, there is no significant difference between the compression ratios achieved by $k^2$ trees and OFR. In addition, we point out future directions for research. All code and datasets are available at https://github.com/dice-group/GraphCompression and https://hobbitdata.informatik.uni-leipzig.de/rdfrepair/evaluation_datasets/, respectively.

**Keywords:** RDF compression· Graph compression · Benchmarking.

## 1 Introduction

The first prominent use of data compression can be traced back to the 19th century with works such as the Morse code, which uses a precursor of entropy-based compression by assigning shorter codes to high-frequency letters [16]. Data compression is now used in an ever growing number of settings, especially due to the steadily increasing size of application-relevant datasets [7,8,10,12,15,17]. This holds in particular for RDF knowledge graphs, which grow continuously in both number and sheer size [10]. The need for compressing RDF data has hence fueled a considerable body of research.

RDF compression algorithms achieve better compression ratios than human-readable compact RDF representations (e.g., Turtle [3], Notation-3 [4]) by serializing RDF data in a manner which still allows for querying. The wide range of available approaches spans algorithms implemented directly in storage solutions [2] over algorithms able to exploit the semantics of RDF knowledge graphs [12,15] to syntax-based compression

techniques [8,10]. Most of these approaches abide by the general concept of separating an RDF graph into three different parts [8]: a header, a dictionary and a representation of the triples. The *header* contains general statistical information. Since it is not necessary for the decompression of the RDF graph, we will not further take it into consideration throughout the rest of this paper. The *dictionary* maps the URIs and literal values of the graph to ids. These ids are used within the triples file for a space-efficient *representation*. This general concept is wide spread when it comes to the compression of RDF graphs [8,10].

The graph processing community has also been aware of the need for compression and has developed a range of approaches ranging from tree-based strategies [5] to techniques based on automatically generated graph context-free grammars [13]. Especially the latter work of Maneth et al. attracted our interest since the authors implemented a prototypical compressor and evaluated it on different types of graphs (including some RDF graphs). Based on a comparison with a $k^2$-trees-based compression they conclude that "[o]ver RDF graphs [...] our compressor gives the best results, sometimes factors of several magnitudes smaller than other compressors" [13]. However, no previous work has addressed the concrete task of porting and comparing the current state of the art in grammar-based graph compression with the current reigning RDF compression algorithms. We address exactly this research gap.

In this paper, we port one of the currently best performing graph compression approaches, i.e., gRePair [13] and adapt it to RDF knowledge graphs. In addition, we develop an efficient implementation of $k^2$ trees [5,1] for RDF. The resulting approaches are compared with HDT, HDT++ and OFR in a large-scale evaluation over 40 datasets w.r.t. their runtime and compression ratio. Our results suggest that OFR and $k^2$ trees achieve comparable results and outperform other RDF compression approaches significantly with respect to compression ratio—including gRePair. Our result analysis unveils more efficient dictionary compression approaches yield the potential for better RDF compression ratios. A comparison with respect to query execution performance is not part of this paper.

In the following Section, we present related work. Section 3 comprises preliminaries before our approach is described in Section 4. We describe our evaluation and report results in Section 5 before we conclude in Section 6.

## 2   Related work

The existing compression algorithms for RDF data can be separated into two groups— syntactic compression algorithms and semantic compression algorithms. A syntactic compression takes the given RDF graph and uses an economical syntax to encode its information. For example, Fernández et al. [8] present the HDT compression.[3] It is an implementation of a dictionary and several triple representations. The dictionary reduces the space needed to store the URIs by using a prefix tree. The most efficient triple representation groups triples by their subject and after that by their predicate. The grouped triples are represented by using id arrays and bitsets. Álvarez-García et al. [2,1]

---

[3] https://www.w3.org/Submission/HDT/

the $k^2$-triples approach that uses $k^2$ trees to store triples to ensure that even large RDF graphs can be handled by in-memory data stores. However, the authors do not compare the approach with other RDF graph compression approaches. Similar to $k^2$-triples, Wang et al. [18] proposes the usage of octrees to compress the representation of triples. Their evaluation shows that this approach achieves better compression ratios than HDT for 4 example datasets. However, to the best of our knowledge the implementations of $k^2$-triples and octrees are not publicly available. Hernández-Illera et al. [10] extend HDT to HDT++ by using predicate families, i.e., combinations of predicates that co-occur very often. Instead of storing the predicate ids for each of these triples, HDT++ stores the id of the predicate family together with the object ids. The evaluation shows that especially for highly structured datasets, HDT++ achieves better compression ratios than HDT. For 3 out of the 4 datasets used for the evaluation, HDT++ outperforms a $k^2$-tree-based compression. The Objects-First Representation (OFR) presented by Swacha et al. [17] uses a two-staged algorithm. In the first stage, the dictionary and the triples are compressed. Instead of a single dictionary, the algorithm uses several indexes that handle different parts like subject, predicate or object URIs, subject or object names, or literals. The triples are represented as `object, subject, predicate` tuples and subsequently sorted in ascending order, thus allowing the usage of a delta encoding for the objects. This means that only one bit is necessary to encode whether the object of a triple remains the same as the object of the previous triple or whether its ID is increased by 1. The encoding of the subject follows a similar idea with a special handling of large deltas between the IDs. The predicate IDs are encoded as usual numbers. In the second stage of the algorithm, a general compression algorithm is applied. Depending on the data, the first step uses different output streams to write the data. The authors argue that this allows the second-stage algorithm to find more patterns within streams that contain similar data. In their evaluation, the authors show that using either the Deflate or the LZMA algorithm in the second stage outperforms the HDT algorithm using the same algorithms as a post processing. However, the usage of general compression algorithms prevents the execution of queries on the compressed dataset.

The group of semantic compression algorithms aims at the reduction of the number of triples that need to be stored by replacing repetitive parts of the graph. A general approach to the reduction of graphs are grammar-based compressions. Maneth et al. [13] propose the gRePair algorithm. The approach searches for edge pairs—named digrams—that occur often within a graph. The occurrences of the digrams are replaced by hyper edges. In an additional grammar, the rules for replacing the hyper edges with their digrams is stored. The remaining graph is stored using a $k^2$ tree. Although the authors suggest that this compression can potentially be ported to RDF and used for querying, we are the first to port gRePair to RDF knowledge graphs. Pan et al. [15] propose to search for redundant graph patterns and replace them by triples with newly created predicates and a grammar comprising rules for the decompression. However, a comparison with existing RDF compression approaches like HDT with respect to their compression ratio is missing and the source code is not available. Gayathri et al. [9] propose the mining of logical Horn-rules. Based on these rules, triples that can be inferred are removed from the graph. In the evaluation, the authors show that depending on the dataset 27–40% of the triples can be removed. In a similar way, Joshi et al. [12] propose

a compression technique which is based on frequent item set mining. Frequent patterns that can be recovered by applying rules are removed from the graph. Their evaluation shows that for several datasets, more than 50% of the triples can be removed and that the removal can lead to an improvement of the performance of the HDT compression algorithm.

## 3   Preliminaries

**Definition 1 (Sets).** *Let $U$, $B$ and $L$ be the mutually disjoint sets of URI references, blank nodes and literals, respectively [11]. Let $P \subseteq U$ be the set of all properties.*

**Definition 2 (RDF triple).** *An RDF triple $t = (s, p, o) \in (U \cup B) \times P \times (U \cup B \cup L)$ displays the statement that the subject $s$ is related to the object $o$ via the predicate $p$ [11].*

**Definition 3 (RDF graph).** *An RDF graph can be defined as directed labeled multi-graph, i.e., as a tuple $G = (V, E, \lambda)$ where $V = \{v_1, ..., v_n\}$ is the set of nodes; $E$ is a multiset of edges $e_i = (v_i^{(t)}, v_i^{(h)}) \in V^2$ and $\lambda : E \to P$ is the edge label mapping.*

**Definition 4 (Digram).** *A digram $d = (p_i, p_j)$ is defined as two edges that share at least one node and are labeled with two edge labels $p_i$ and $p_j$. It follows, that each digram can link up to three nodes. Each node is either an external or an internal node, where a node is called external if it has at least one edge that does not belong to the digram.*

Note that in contrast to Maneth et al. [13], we do not define digrams as hyperedges, i.e., we limit ourselves to digrams with one or two external nodes for two reasons: First, this allows the usage of digrams as normal edges in a directed labeled graph as defined above. Second, preliminary implementations showed that digrams with more than two external nodes might not lead to better compression ratios. Our definition leads to 33 different shapes of digrams. 8 examples thereof are depicted in Figure 1. A digram occurrence is defined as the occurrence of such a digram in a given graph.

**Definition 5 (Digram-compressed RDF graph).** *Let $D$ be the set of all digrams. A digram-compressed RDF graph is an RDF graph which has an extended label mapping function $\lambda' : E \to P \cup D$.*

**Definition 6 (Non-terminal edge).** *A non-terminal edge is an edge in a digram-compressed RDF graph that is not mapped to a predicate but to a digram.*

**Definition 7 (Grammar).** *A grammar $\mathfrak{G}$ is defined as $\mathfrak{G} = (S, D)$, where $S$ is a digram-compressed RDF graph named* start graph *and $D$ is the set of digrams used to compress the graph.*

**Definition 8 (Quadrant).** *A matrix of dimension $2^n \times 2^n$ can be divided into four sub-matrices of equal size $2^{n-1} \times 2^{n-1}$. These submatrices are called quadrants. The quadrants will represent the following rows and columns of the original matrix:*

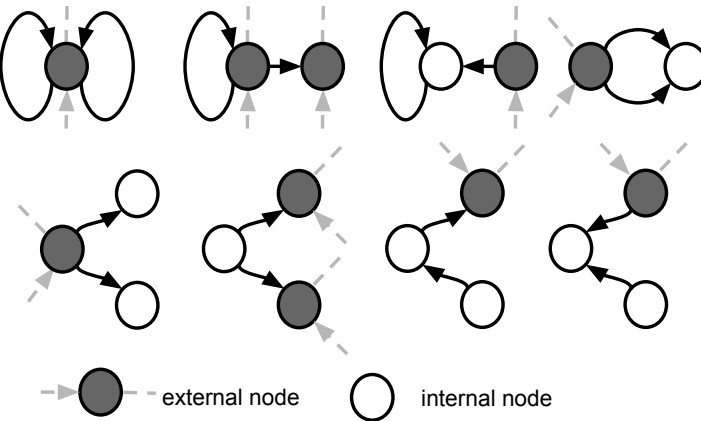

Fig. 1: Examples of digram shapes. The external nodes have additional edges that are not part of the digram (depicted in light gray).

- *First quadrant: (0, 0) to $(2^{n-1} - 1, 2^{n-1} - 1)$;*
- *Second quadrant: $(0, 2^{n-1})$ to $(2^{n-1} - 1, 2^n)$;*
- *Third quadrant: $(2^{n-1}, 0)$ to $(2^n, 2^{n-1} - 1)$;*
- *Fourth quadrant: $(2^{n-1}, 2^{n-1})$ to $(2^n, 2^n)$.*

**Definition 9  (Compression Ratio).** *Let $s_o$ and $s_c$ be the file size in bytes of the original RDF file and the compressed file, respectively. The compression ratio $r$ is defined as $r = \frac{s_c}{s_o}$. The smaller the compression ratio, the better is the performance of a compression algorithm.*

## 4  Approaches

We implemented two approaches for RDF compression: RDFRePair and $k^2$. We begin by presenting RDFRePair, an RDF compression approach based on the gRePair algorithm proposed by Maneth et al. [13]. It adapts the gRePair approach to RDF and combines it with the dictionary of [8]. The workflow of RDFRePair comprises 4 main steps: 1) Indexing the nodes and edge labels of the input graph, 2) running the gRePair algorithm, 3) creating $k^2$ trees for the remaining, compressed graph and 4) serializing the graph. The second approach skips the second step and solely relies on an efficient implementation of $k^2$ trees as proposed by Álvarez-García et al. [1]. These steps are explained in the following before we explain the decompression of the graph in 4.5. The execution of queries on the compressed graph is out of the scope of this paper.

### 4.1  Indexing

The first step is to load the input RDF graph into memory and index all nodes and edge labels within the graph. Maneth et al. [13] state that "[a]ny method for dictionary compression can be used to additionally compress the dictionary (e.g. [14])". Hence, we use the dictionary implementation of HDT [8].

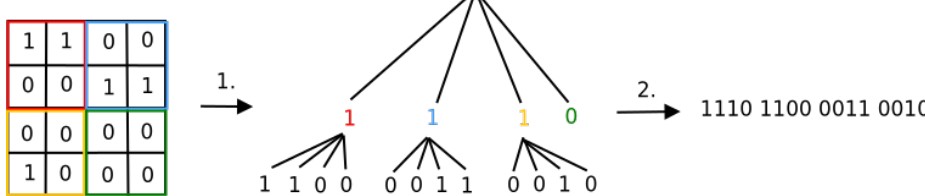

Fig. 2: Example of an adjacency matrix to a $k^2$ tree and its serialization.

### 4.2 gRePair Algorithm

In this step, the algorithm will create a Grammar $\mathfrak{G}$ from the indexed graph $G$ as described by Maneth et al. [13]. This step consists of the following sub steps.

1. *Initial digram scan:* The algorithm iterates over all vertices. For each vertex, all pairs of edges connected to that vertex are counted as potential digrams.
2. *Sort digrams:* All digrams with at least two occurrences are sorted descending by their frequency using a priority queue.
3. *Get most frequent digram $d$:* The most frequent digram is removed from the priority queue.
4. *Replace all occurences of $d$:* All occurrences of $d$ within the graph are replaced with non-terminal edges and the edges receive the label $d$. All replaced occurrences are added to a list which is necessary for the later serialization of the digram $d$.
5. *Find new digrams:* Since new edges have been introduced, new digrams could have been created. All vertices connected to at least one of these newly created non-terminal edges are given to the digram search algorithm to search for new digrams. If new digrams are found, they are added to the queue.
6. *Repeat:* if the queue is not empty, go back to step 2.

### 4.3 $k^2$ Trees

The grammar $\mathfrak{G}$ created by the gRePair algorithm is split up in a start graph $S$ and the set of digrams $D$. As proposed by Maneth et al. [13], an adjacency matrix is created for each edge label in $S$. The matrix is of dimension $|V| \times |V|$ and its cells represent the edges between the subject (row index) and the object (column index). If an edge with the edge label of the matrix exists between a subject and an object the representing cell is set to 1. Hence, the matrix is typically sparse.

Thereafter, the $k^2$ trees are built from these matrices. To this end, each path from the root of the $k^2$ tree to its leaves is built individually before it is merged with all other paths of the matrix. The path creation algorithm is shown in Algorithm 1. First, the matrix is resized to $2^h \times 2^h$ where $h \in \mathbb{N}$ is the lowest integer that fulfills $2^h \geq |V|$. The added rows and columns are filled with zeros. After that, the matrix is transformed into a tree using a recursion. Starting with the root node, the matrix is divided into 4 quadrants as defined in Definition 8 and four child nodes are added to the root node. The value of a child node is either 1 if the quadrant contains at least one cell with a

---

**Algorithm 1:** $k^2$ tree path creation algorithm.

---

**Input:** Matrix M, Integer h
**Output:** $k^2$-Tree
1  x1 = 0, y1 = 0, x2 = $2^h$, y2 = $2^h$
2  root = new TreeNode()
3  currentNode = root
4  **for** *Point p : M.getPoints()* **do**
5  |    quadrant = getQuadrant(p, x1, y1, x2, y2)
6  |    child = new TreeNode()
7  |    currentNode.set(quadrant, child)
8  |    currentNode = child
9  |    shrinkBoundaries(x1, y1, x2, y2, quadrant)
10 **return** *root*

---

---

**Algorithm 2:** $k^2$ tree path merge algorithm.

---

**Input:** TreeNode node, Map<Integer, TreeNode> map, Integer k, Integer h
**Output:** List of individual paths in $k^2$-tree
1  **if** *k==h OR node == null* **then**
2  |    return
3  **for** *child C : node.getChildren()* **do**
4  |    map.get(k).add(C)
5  **for** *child C : node.getChildren()* **do**
6  |    merge(C, map, k+1, h)
7  **return** *map*

---

1 value. Otherwise, the child node gets the value 0. This is done recursively for each child having a 1 until the quadrants are of size 2x2. In that case, the 4 numbers of the quadrant are used for the 4 child nodes.

Instead of implementing the recursion directly, we implemented a more efficient algorithm, which comprises two steps. First, the algorithm iterates over all cells of the matrix having a 1 value. For each of these cells, the path within a $k^2$ tree is determined as shown in Algorithm 1. Beginning with the complete matrix, the algorithm determines the quadrant in which the cell is located and adds a child node to the path before shrinking the quadrant. Thereafter, all generated paths are merged as shown in Algorithm 2. Afterwards the map represents the $k^2$ Tree optimized for later serialization.

### 4.4   Serialization

The serializiation of the created grammar $\mathfrak{G}$ comprises the serialization of the start graph and the serialization of the digrams.

*Start graph.* The start graph is serialized as a sequence of its $k^2$-trees. Each tree is preceded by the ID of its edge label (4 bytes). Each tree node is represented by a single

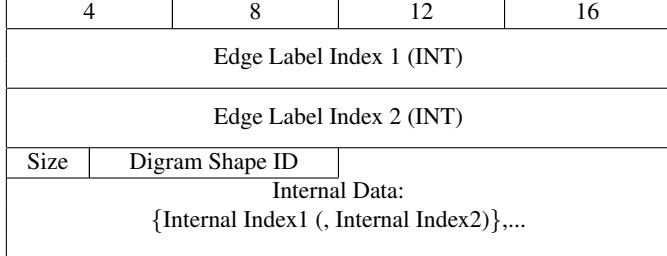

| 4 | 8 | 12 | 16 |
|---|---|---|---|
| Edge Label Index 1 (INT) | | | |
| Edge Label Index 2 (INT) | | | |

| Size | Digram Shape ID | | |
|---|---|---|---|
| Internal Data:
{Internal Index1 (, Internal Index2)},... | | | |

Fig. 3: Digram serialization. One line represents 2 bytes.

bit. Hence, the tree is serialized as a sequence of bits representing its nodes from top to bottom. If the tree has an uneven number of nodes, the last byte will be padded with zeros. An example is shown in Figure 2. In this example the whole tree can be stored using only 2 bytes.

*Digrams.* The digrams used to reduce the graph are serialized as depicted in Figure 3. A serialized digram comprises the two edge labels, a size flag, a shape ID and the IDs of all internal nodes of all occurrences of this digram. The edge label IDs are two integers that represent the IDs of the properties or digrams the two edges of the digram have.[4] The size flag uses two bits to decode the number of bytes that are used for the single internal node IDs. This allows the usage of 1, 2, 3 or 4 byte IDs. The shape ID comprises 6 bits that are used to store the ID of the digram shape (i.e., one of the different 33 shapes). The last part of the digram lists the IDs of the internal nodes of all occurrences of the digram. To this end, the occurrences of the digram are sorted based on the ID(s) of its external nodes. Hence, the mapping of internal nodes to the single occurrences of the digram are stored implicitly without taking any additional space. Maneth et al. [13] propose an optimization that reassigns the IDs of vertices in the graph to implicitly store the IDs of internal nodes as well. However, this would raise a new requirement for the dictionary or consume additional space to store the mapping of IDs.

### 4.5  Decompression

In this section, we briefly describe the process of decompressing a compressed graph. First, the dictionary is loaded. After that all $k^2$ trees for all terminal edges are loaded and directly transformed into the RDF triples they represent. The digrams are then read and iterated upon in reverse order. The non-terminal edges of each digram's $k^2$-tree are sorted by the IDs of vertices they connect. Based on this order, the single non-terminals can be replaced by the two edges and the internal nodes. Since the non-terminal edges are handled in the same order as during the serialisation, the internal nodes are read in the correct order. Depending on the two edge labels a digram contains, the generated

---

[4] In the current implementation, we use 32 Bit integers. They can be extended to 64 Bits for very large graphs.

terminal edges are directly written into the result RDF graph. If non-terminals are created, they are added to the list of their digram. The order of digrams ensures that only non-terminal edges of yet unprocessed digrams can be found.

## 5 Evaluation

Our evaluation aims to answer the following research questions:

- **RQ1**: How do RDFRePair and $k^2$ perform compared to state-of-the-art RDF graph compression algorithms w.r.t. compression ratio and (de)compression times?
- **RQ2**: To which extent does the dictionary affect the compressed size?
- **RQ3**: Which RDF dataset features influence the compression ratio?

### 5.1 Experimental Setup

To answer the research questions above, we execute several experiments using different RDF datasets and compression algorithms.[5] For each dataset-algorithm combination, we use the algorithm to compress and decompress the dataset. During that, we gather four measures: 1) the compression ratio, 2) the runtime of the compression, 3) the runtime of the complete decompression and 4) the amount of space of the compressed dataset that is used to store the dictionary. In addition, we analyze the datasets using the following metrics to answer the third question: number of triples, classes and resources, URI resources, properties, as well as star pattern similarity and structuredness. We elaborate upon the last two measures in the following.

In [13], the authors mention that a graph similar to the star pattern is beneficial for gRePair, because gRePair can make use of this structure to find many digram occurrences around those high-degree nodes. A directed graph is described as a star pattern if one node $v \in V$ is connected to all other nodes, whereas no other nodes are connected to each other. Hence, the following necessary (but not sufficient) condition must apply: $\exists v \in V, \forall e \in E : v \in e$. As graphs tend to be more complex than such simple patterns, we define a metric describing how similar a given graph is to a star pattern. Let $deg(v)$ be the degree of a node $v$. Let $N$ be a list of all nodes sorted by their deg-values in descending order and $N_x$ be the first $x$ nodes of $N$. We define the star pattern similarity ($SPS$) metric as follows.

$$SPS = \frac{\sum_{n \in N_x} deg(n)}{\sum_{n \in N} deg(n)} \in [0 : 1] \tag{1}$$

In our experiments we choose $x = 0.001 \cdot |N|$.

Duan et al. [6] compare synthetic and real-world RDF datasets and conclude that synthetic datasets tend to be more structured. To measure this structuredness, they count how regularly properties occur for instances of classes. If all instances of a class have

---

[5] All experiments were executed on a 64-bit Ubuntu 16.04 machine, an Intel(R) Xeon(R) CPU E5-2698 v3 @ 2.30GHz with 64 CPUs and 128GB RAM. Only the experiments for WatDiv were executed on a 64-bit Debian machine with 128 CPUs and 1TB RAM.

Table 1: Datasets used for the evaluation.

| Name | Abbreviation | #Triples | #Resources | #Classes |
|------|-------------|---------|-----------|---------|
| dc-2010-complete-alignments | SD0 | 5 919 | 821 | 36 |
| ekaw-2012-complete-alignments | SD1 | 13 114 | 1 604 | 36 |
| eswc-2006-complete-alignments | SD2 | 6 654 | 1 259 | 25 |
| eswc-2009-complete-alignments | SD3 | 9 456 | 1 247 | 34 |
| eswc-2010-complete-alignments | SD4 | 18 122 | 2 226 | 36 |
| eswc-2011-complete-alignments | SD5 | 25 865 | 3 071 | 36 |
| iswc-2002-complete-alignments | SD6 | 13 450 | 1 953 | 36 |
| iswc-2003-complete-alignments | SD7 | 18 039 | 2 565 | 36 |
| iswc-2005-complete-alignments | SD8 | 28 149 | 3 877 | 36 |
| iswc-2010-complete-alignments | SD9 | 32 022 | 3 842 | 36 |
| external_links_en | DB0 | 49 999 | 7 070 | 0 |
| geo_coordinates_en | DB1 | 49 999 | 54 870 | 1 |
| homepages_en | DB2 | 49 999 | 12 505 | 0 |
| instance_types_transitive_en | DB3 | 49 999 | 98 666 | 273 |
| instance_types_en | DB4 | 49 999 | 48 913 | 306 |
| mappingbased_objects_en | DB5 | 49 998 | 37 159 | 0 |
| persondata_en | DB6 | 49 999 | 9 516 | 2 |
| transitive_redirects_en | DB7 | 49 999 | 82 386 | 0 |
| wikidata-20200308-lexemes-BETA | WD0 | 49 828 | 9 965 | 15 |
| wikidata-20200404-lexemes-BETA | WD1 | 49 827 | 9 931 | 15 |
| wikidata-20200412-lexemes-BETA | WD2 | 49 828 | 9 932 | 15 |
| wikidata-20200418-lexemes-BETA | WD3 | 49 828 | 9 902 | 15 |
| lubm-1 | LUBM-1 | 100 545 | 17 209 | 15 |
| lubm-10 | LUBM-10 | 1 272 577 | 207 461 | 15 |
| lubm-100 | LUBM-100 | 13 405 383 | 2 179 801 | 15 |
| lubm-1000 | LUBM-1000 | 133 573 856 | 21 715 143 | 15 |
| watdiv | WAT | 1 098 871 666 | 52 120 471 | 12 500 145 |
| external_links_en | EL | 7 772 283 | 9 128 582 | 0 |
| geo_coordinates_en | GC | 2 323 568 | 580 897 | 1 |
| homepages_en | HO | 688 563 | 1 300 927 | 0 |
| instance_types_en | IT | 5 150 432 | 5 044 646 | 422 |
| instance_types_transitive_en | ITT | 31 254 270 | 4 737 461 | 388 |
| mappingbased_objects_en | MO | 18 746 173 | 5 901 219 | 0 |
| persondata_en | PD | 10 310 094 | 1 522 938 | 18 |
| transitive_redirects_en | TR | 7 632 358 | 10 404 804 | 0 |
| archives-hub | AH | 1 361 815 | 135 643 | 46 |
| jamendo | JA | 1 047 950 | 410 929 | 11 |
| scholarydata_dump | SDD | 859 840 | 95 016 | 46 |
| wikidata-20200308-lexemes-BETA | WD | 42 914 845 | 6 061 049 | 22 |
| dblp-20170124 | DBLP | 88 150 324 | 28 058 722 | 14 |

triples with the same properties, the class is highly structured. If some of the instances have triples with properties that other instances of the same class do not have, the class is less structured. The structuredness of a dataset is the weighted average of the class

structuredness values with a higher weight for classes with many instances and many properties.

## 5.2   Datasets

Table 1 shows the summary of the datasets.[6] We use 40 datasets in total. Note that gRe-Pair and in part RDFRePair were not able to handle large datasets in our experiments. To still be able to compare them with other approaches, we introduce 4 Wikidata and 8 DBpedia subsets cut at 50k lines.

## 5.3   Compression Algorithms

For the evaluation of RDFRePair, we select a subset of the algorithms listed as related work. We choose HDT because of its wide adoption and its usage as reference algorithm in several publications.[7] In addition, we compare our evaluation with HDT++ and OFR since both algorithms are reported to perform at state-of-the-art level and better than HDT.[8] We also use our $k^2$ implementation since the implementations used in the related work (e.g., [1]) do not seem to be available as open source. In addition, we received a prototypical implementation of the gRePair algorithm from the authors of [13]. Given that the original gRePair implementation is a proof of concept, it is rather far from stable regarding decompression. Apart from that, the implementation does not create a dictionary. To alleviate this problem, the HDT dictionary size was added for a fair comparison. The OFR compression provides several files representing the compressed graph. To combine these files and further compress them the authors suggested to use either Deflate (zip) or LZMA (7z). However, since our goal is to compare comparisons that would be able to answer SPARQL queries, we do not use additional, binary compression algorithms. Instead, we sum up the sizes of the individual files. The addition of other algorithms (see Section 2) was prevented by the non-availability of their implementation or their reported poor compression ratio.

## 5.4   Results

Figure 4 shows the compression ratios achieved by the different algorithms. To compare the compression ratios across the different datasets, we use a one-tailed Wilcoxon signed-rank test for a pairwise comparison of the compression algorithms. Table 2 lists the p-values of the tests. These results suggest that RDFRePair leads to significantly better compression ratios than the original prototypical gRePair implementation. However, RDFRePair is significantly outperformed by $k^2$ and OFR. The prototypical implementation of gRePair is outperformed by all other approaches. Overall, OFR and our implementation of the $k^2$ algorithm lead to the best compression ratios with $k^2$ performing

---

[6] The datasets can be found at https://w3id.org/dice-research/data/rdfrepair/evaluation_datasets/. For scholarly data (DF0–DF9), we use the rich datasets (see http://www.scholarlydata.org/dumps/).

[7] https://github.com/rdfhdt/hdt-java

[8] HDT++ is available at https://github.com/antonioillera/iHDTpp-src. OFR is not publicly available. However, the authors were so kind to provide us the binaries.

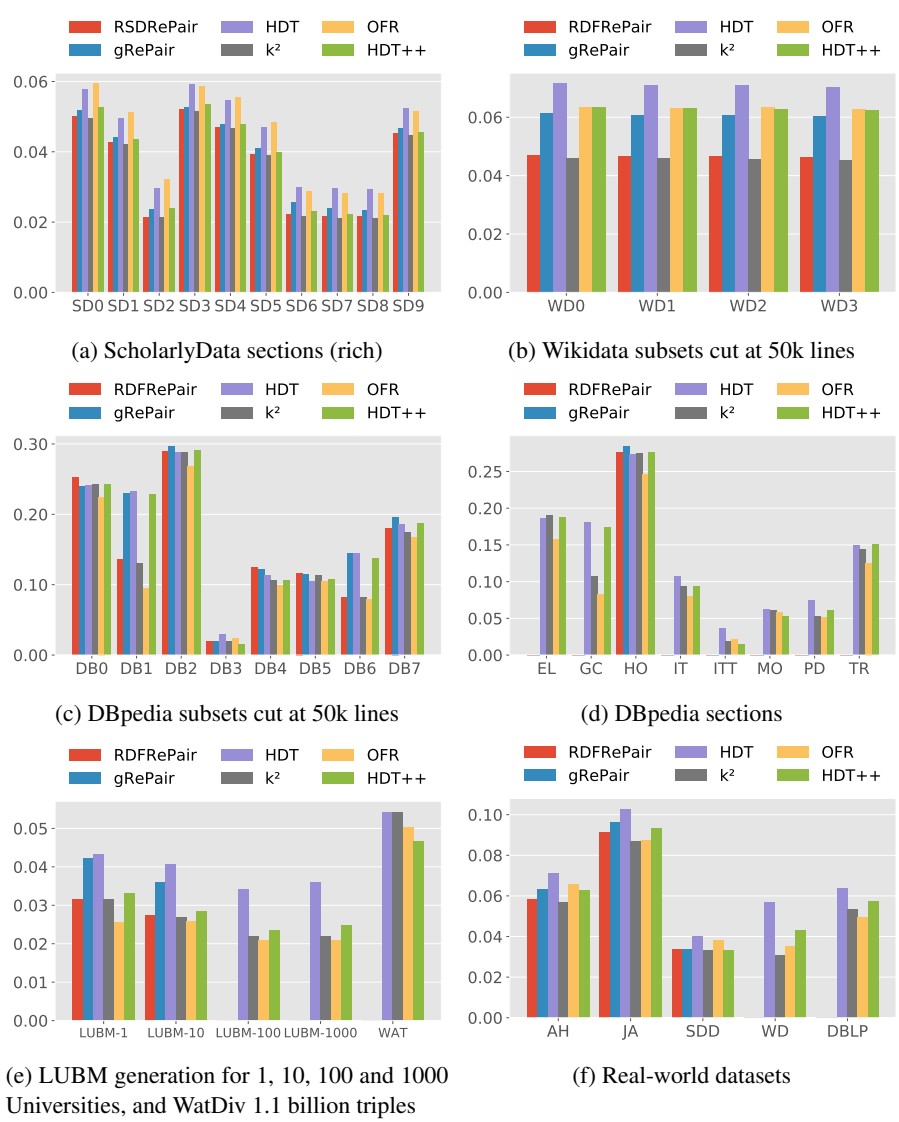

(a) ScholarlyData sections (rich)

(b) Wikidata subsets cut at 50k lines

(c) DBpedia subsets cut at 50k lines

(d) DBpedia sections

(e) LUBM generation for 1, 10, 100 and 1000
Universities, and WatDiv 1.1 billion triples

(f) Real-world datasets

Fig. 4: Compression ratio for the compression algorithms on the single datasets. Smaller values are better.

Table 2: $p$-value of a one-tailed Wilcoxon signed rank test with respect to compression ratio. A bold value indicates that the algorithm in the row leads to a significantly better compression ratio than the algorithm in the column ($p < \alpha = 0.05$).

| $r_1 \setminus r_2$ | RDFRePair | gRePair | HDT | $k^2$ | OFR | HDT++ |
|---|---|---|---|---|---|---|
| RDFRePair | — | $\approx$**0.0** | 0.79 | $\approx$1.0 | 0.99 | 0.89 |
| gRePair | $\approx$1.0 | — | 0.95 | $\approx$1.0 | $\approx$1.0 | $\approx$1.0 |
| HDT | 0.21 | 0.05 | — | $\approx$1.0 | $\approx$1.0 | $\approx$1.0 |
| $k^2$ | $\approx$**0.0** | $\approx$**0.0** | $\approx$**0.0** | — | 0.51 | $\approx$**0.0** |
| OFR | **0.01** | $\approx$**0.0** | $\approx$**0.0** | 0.49 | — | 0.07 |
| HDT++ | 0.11 | $\approx$**0.0** | $\approx$**0.0** | $\approx$1.0 | 0.93 | — |

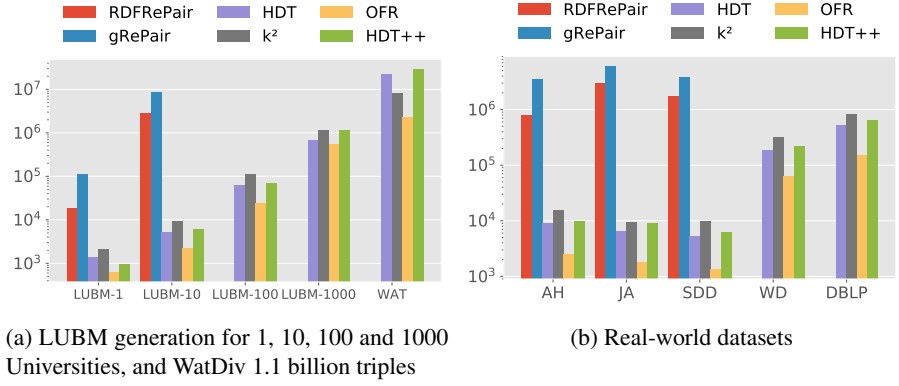

(a) LUBM generation for 1, 10, 100 and 1000 Universities, and WatDiv 1.1 billion triples

(b) Real-world datasets

Fig. 5: compression time in ms (log). Smaller values are better.

better on non-synthetic datasets. None of these two algorithms is able to significantly outperform the other one with respect to the compression ratio. Our findings contradict the results of Maneth et al. [13] that gRePair performs better than $k^2$.

OFR has the shortest runtime w.r.t. compression and decompression time (depicted in Figures 5 and 6). It is followed by HDT and HDT++. $k^2$ shows a longer runtime than these three algorithms. The prototypical implementation of gRePair is the slowest algorithm. In addition, gRePair and RDFRePair were not able to compress 12 of the 40 datasets within 2 hours, respectively.

Figure 7 depicts the amount of space used to store the compressed dictionary in comparison to the overall size of the compressed dataset. 5 of the 6 approaches share a similar dictionary implementation based on [8]. For all these approaches, the dictionary consumes the majority of the space. Especially for the $k^2$ compression, the average size of the dictionary over all datasets is 80%. In comparison, the OFR dictionary achieves smaller dictionary sizes on some of the datasets. This suggests that improvements to the dictionary can lead to much better compression ratios for HDT, HDT++ and $k^2$.

The correlation analysis reveals that all algorithms have a correlation between their performance and the number of classes. However, this seems to be an indirect relation

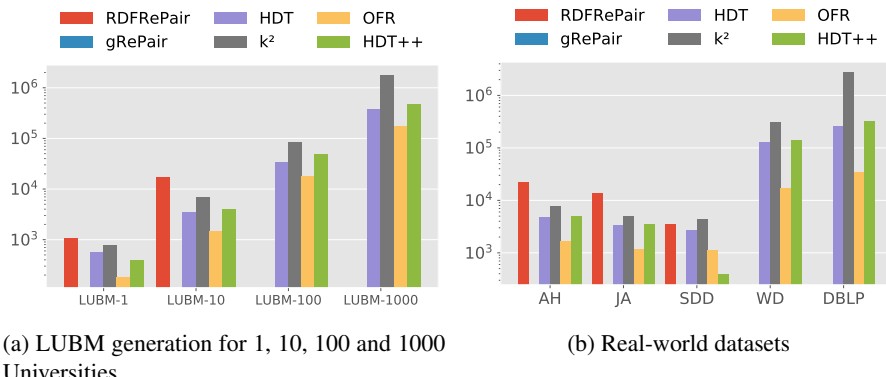

(a) LUBM generation for 1, 10, 100 and 1000 Universities

(b) Real-world datasets

Fig. 6: decompression time in ms (log). Smaller values are better.

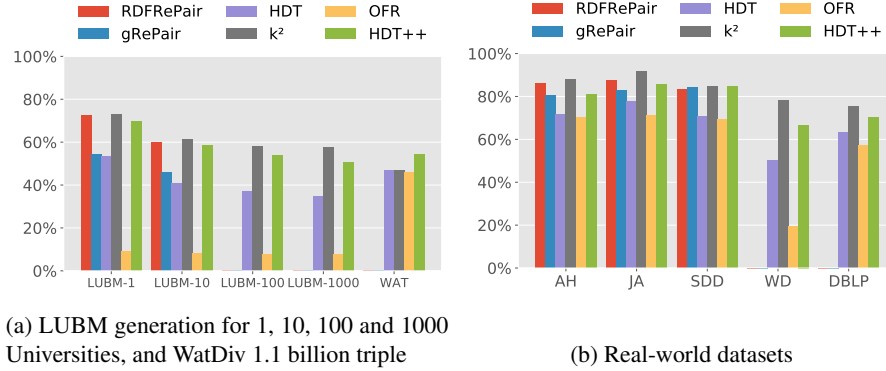

(a) LUBM generation for 1, 10, 100 and 1000 Universities, and WatDiv 1.1 billion triple

(b) Real-world datasets

Fig. 7: Size of the dictionary in comparison to the compressed dataset size in %.

that is caused by the datasets that contain solely one-to-one mappings like DB0 or EL. These datasets have no classes and are hard to compress because of their one-to-one structure. Neither the SPS nor the structuredness metric show any significant correlations.

The results show that RDFRePair underachieves. A further analysis reveals that the optimization described in Section 4.4, i.e., to store the IDs of internal nodes implicitly by renumbering all nodes in the graph, is one of the major features of gRePair.[9] This leads to very good results in the evaluation done by Maneth et al. [13]. This optimization seems to contradict the statement that a dictionary compression as proposed by Martínez-Prieto et al. [14] can be used (see Section 4.1) since the dictionary compression needs to allow the gRePair algorithm to freely redefine the IDs of all nodes. However, Martínez-Prieto et al. separate the space of node IDs into several ranges. Based on

---

[9] For a fair comparison, we turned this feature of gRePair in our evaluation off. Otherwise, it couldn't be used with the HDT dictionary.

Table 3: Values of Kendall's Tau rank correlation between compression ratio and dataset metrics. A bold value indicates a significant correlation ($\alpha = 0.02$). * only experiments that terminated in time were taken into account.

|  | RDFRePair* | gRePair* | HDT | $k^2$ | OFR | HDT++ |
|---|---|---|---|---|---|---|
| #Triples | 0.27 | 0.32 | 0.17 | 0.13 | 0.03 | 0.11 |
| #Classes | **−0.47** | **−0.57** | **−0.51** | **−0.50** | **−0.43** | **−0.52** |
| #UriResources | 0.23 | 0.27 | 0.17 | 0.13 | 0.04 | 0.10 |
| #Resources | **0.37** | **0.40** | 0.24 | 0.20 | 0.12 | 0.17 |
| #Properties | −0.17 | −0.13 | −0.22 | −0.24 | −0.15 | −0.19 |
| SPS | 0.09 | 0.14 | 0.06 | 0.00 | −0.01 | 0.04 |
| Structuredness | −0.20 | −0.24 | −0.21 | −0.18 | −0.30 | −0.21 |

the role of a node in the graph, it has to receive an ID of a certain range. This allows different indexing and compression strategies for the different ranges. However, such a node can not get an ID assigned by gRePair. We measured the amount of space the internal nodes consume. Especially for type graphs with a simple structure (e.g., DB4) the internal nodes consume up to 98% of the memory of the compressed triples (i.e., the memory of the compressed dataset without the dictionary). We call this the *internal node size ratio*. The datasets used by Maneth et al. [13] seem to favor the usage of digrams. Two out of the six datasets have an internal node size ratio of 99% while three other datasets have ratios of more than 50%. In our evaluation, the majority of datasets has an internal node size ratio below 50%. More diverse datasets like SD0–SD9 have ratios between 9% and 16%. Even without storing internal nodes, RDFRePair and gRePair would still show a lower performance than $k^2$ trees for such datasets but with the cost of a dictionary that may less optimized for querying and compression.

## 6  Conclusion

This paper presented several contributions. First, we presented RDFRePair—an improved implementation of the gRePair algorithm ported to the compression of RDF graphs. Second, we present an efficient implementation of the $k^2$ trees for the same goal. Third, we ran a large-scale evaluation comparing RDFRePair and $k^2$ with HDT, HDT++, OFR and a prototypical implementation of gRePair. Our results could not support the assumption that grammar-based compressions like gRePair are able to outperform existing RDF graph compressions. Instead, our results suggest that in most cases the best compression ratio of RDF datasets can be achieved by using either $k^2$ trees or OFR with $k^2$ trees performing best on average when faced with real data. On the other hand, OFR clearly shows a better runtime performance. There are existing implementations for executing SPARQL queries on $k^2$ trees while an implementation for the query execution for OFR is missing. During the analysis of our results, we couldn't identify significant correlations between the compressor's performance and the features of the datasets. However, our results suggest that future work may focus on further improving the dictionary since it consumes the majority of the space.

## Acknowledgements

This work has been supported by the German Federal Ministry for Economic Affairs and Energy (BMWi) within the project SPEAKER under the grant no 01MK20011U and by the EU H2020 Marie Skłodowska-Curie project KnowGraphs under the grant agreement no 860801.

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
