# OpenReview forum: "Applying Grammar-based Compression to RDF"
_eswc-conferences.org/ESWC/2021/Conference/Research_Track — ESWC 2021 Research_

### Official Review · AnonReviewer3 · 2021-01-09
**A grammar-based (syntactic) RDF compressor.**

**Rating:** 1
**Confidence:** 5
**Impact:** 3
**Design And Technical Quality:** 4

**Review:**

This paper adapts the graph-based compressor gRePair to the particular case of RDF graphs. To the best of my knowledge, this technique (called RDFRePair) is the first grammar-based compressor for knowledge graphs and I consider that it is a valuable contribution to the community, since grammar-based compression can give space-time tradeoffs complementary from those achieved by state-of-the-art compressors such HDT or k2-triples. Besides, the authors provide a k2-triples (Álvarez-García, et al. 2015) implementation (called k2) that improves the original one in compression speed. Prototypes of both compressors are publicly available at GitHub, enabling experiment reproducibility.

The paper is well-organized, and it is easy to read for any people, without being specialist in compression. It also provides a valuable overview of RDF compression techniques, although it is missing two important ones [1,2], that should be included in the experimentation to consolidate a more valuable analysis. The preliminaries set clearly and precisely the definitions to understand the current approaches, although it is not clear to me what is “the file size… of the original RDF file” used to obtain compression ratio, because this size depends on the serialization format.

The paper contribution is simple (but relevant) since RDFRePair is a straightforward implementation of gRePair, restricted to digrams with no more than two external vertices. The authors justify this decision in preliminary experiments, but do not show them. I think that it is an interesting result because this restriction prevents, for instance, different property path patterns (larger than 2 external nodes) to be compressed as digrams. The compression algorithm is summarized in Section 4.2. The resulting diagram-compressed graph is then represented using the authors implementation of k2trees. In practice, it is a k2triples encoding of the start graph; i.e. the original graph enhanced with digrams. Finally, the authors describe how digrams are serialized and I have some considerations:

- Why do you use 32 or 64-bit integer to encode edge label? You can save much bits if you serialize these integers in $\lceil \log_2(|P|+|D|)\rceil$ bits (|P| is the number of different predicates and |D| the number of digrams).

- How do internal node IDs are encoded? Maybe delta-encoding can save some bits if you list these ID in increasing order. Anyway, indexing these internal nodes can be really interesting because it would enable RDFRePair to resolve SPARQL triple patterns, and it would be a more interesting result.

A comprehensive evaluation is described in Section 5, although questions RQ2 and RQ3 have been previously studied in [3] and [4], respectively. The authors choose 40 datasets that represent different types of knowledge graphs, although only two are over 100 million triples. It discovers the main weakness of grammar-based compressor: *scalability*. RDFRePair is not able to compress larger datasets, and it is an important drawback because RDF compression excel in the management of Big Semantic Data. As expected, RDFRePair provides competitive compression ratios for small-medium size datasets, but existing approaches like k2triples or HDT++ are better-suited for larger datasets. I have an important consideration regarding compression ratios:

- In page 13, authors say that "our implementation of the k2 algorithm leads to the best compression ratios", but I do not agree. If I have not misunderstood, this implementation does not change k2triples serialization, so compression ratios are not due to k2 but the original approachof k2-triples. If I didn't understand it right, please rewrite k2 to explain it. On the contrary, please remove this statement because the algorithm effectiveness is due to the original k2-triples.

Figure 5 shows another weakness of grammar-based compressors: *poor compression times*. As expected it RDFRepair (and gRepair) are (by large) the slower compression in the comparison, although it is less relevant in a WORM (write once read many) scenario.

Therefore, the current approach lacks of scalability (discouraging its use for Big Semantic Data compression) and is not efficient for compression (discouraging its use for streaming compression), but I think still that it is a valuable contribution since it fills a gap in the current state of the art and opens a new line of research that can result into interesting RDF self-indexes.

Finally, I suggest the authors to consider this paper [5] if they are interested on improving HDT dictionary. Note that HDT uses Plain Front Coding to compress both URI and Literal dictionaries, but it is only effective for the first one. For Literals, grammar-based dictionaries, like HTFC-rp or HashDAC-rp, that removes much literals redundancy and allows string-to-id and id-to-string transformations to be efficiently performed in compressed space.

[1] Nieves R. Brisaboa, Ana Cerdeira-Pena, Antonio Fariña, Gonzalo Navarro: A Compact RDF Store Using Suffix Arrays. SPIRE 2015: 103-115.

[2] G. E. Pibiri, R. Perego and R. Venturini: Compressed Indexes for Fast Search of Semantic Data. IEEE Transactions on Knowledge and Data Engineering, doi: 10.1109/TKDE.2020.2966609.

[3] Miguel A. Martinez-Prieto, Javier D. Fernández, Rodrigo Cánovas: Compression of RDF dictionaries. SAC 2012: 340-347

[4] Javier D. Fernández, Miguel A. Martínez-Prieto, Pablo de la Fuente Redondo, Claudio Gutiérrez: Characterising RDF data sets. J. Inf. Sci. 44(2): 203-229 (2018)

[5] Miguel A. Martínez-Prieto, Nieves R. Brisaboa, Rodrigo Cánovas, Francisco Claude, Gonzalo Navarro: Practical compressed string dictionaries. Inf. Syst. 56: 73-108 (2016)


**Anonymity:**

No, I would like my review to be deanonymized.

**Reuse And Availability:**

5: Very High

**Strong Points:**

- It proposes the first grammar-based compressor for RDF, filling a gap in the state of the or RDF compression.
- It provides a comprehensive experimentation that compare space-time tradeoffs of many RDF compressors, although it must be enhanced (at least with compressors proposed in [1] and [2]) to make more interesting conclusions.
- Both paper approaches are publicly available at a GitHub repository, making paper research reproducible.

**Subreviewer:**

I submitted this review.

**Weak Points:**

- The paper does not introduce a new approach by itself, because both RDFRePair and k2 are implementations (with a slight variation in RDFRePair and an improved k2tree creation algorithm in k2) of two existing approaches.
- RDFRePair lacks scalability as universal grammar-based compressors. Nevertheless, I think that it can be mitigated if the compression algorithm is performed on a self-indexed representation of the input graph; i.e. indexing the original graph using dynamic k2trees [6]. Thus, RDFRePair would reduce the memory footprint required for compression, because the graph would be indexed in proportional space to that required by k2triples, and it would allow updatings due to selected digrams to be encoded in these k2tree-based encoding.

[6] Nieves R. Brisaboa, Ana Cerdeira-Pena, Guillermo de Bernardo, Gonzalo Navarro: Compressed representation of dynamic binary relations with applications. Inf. Syst. 69: 106-123 (2017)

---

> ### Author Rebuttal · Authors · 2021-01-30
>
> We thank the reviewer for the detailed review.
>
> It is true that we could further save bits in encoding the edge labels in the way proposed by the reviewer. However, we believe that the benefit of this step would have a low impact compared to the effort of its implementation since only 2 edge labels are stored per digram. Hence, we concentrated on the simpler differentiation between 32 and 64 bits. The proposed changes could have been future work in case the compression would have had a better overall performance (not only for the edge labels but especially for the internal node IDs since they consume more space).
> The internal nodes were saved in the order of the external nodes, thus allowing us to decompress the digrams in the same order. Hence, the order of the internal node IDs is fixed and a change of the order would lead to the storage of more data, which would be necessary to map the internal nodes to their external nodes.
> As stated in the related work, k2-trees have been used for RDF graph compression before. The sentence was not meant to claim that we would have been the first to do so. The problem that we had with k2 implementations is that (to the best of our knowledge) there are no open-source implementations for the compression of RDF graphs and, hence, we had to implement it by ourselves. This is why we wrote “our implementation of the k2 algorithm”. However, we understand the concerns of the reviewer regarding the statement and agree to change it so that there is no space for such misunderstandings.

---

> > ### Comment · AnonReviewer3 · 2021-02-02
> > **Rebuttal acknowledgement**
> >
> > Dear authors, thanks for your clarifications, but I have decided to keep my scores. I hope that my questions and suggestions will help you to improve the manuscript.

---

### Official Review · AnonReviewer1 · 2021-01-11

**Rating:** 1
**Confidence:** 4
**Impact:** 3
**Design And Technical Quality:** 4

**Review:**

The authors provide a detailed study of different methods for compressing RDF datasets. To this end, they both implement classical methods (k2 trees), and adapt known graph compression tools (gRePair), in order to compare them with other established solutions for RDF compression (HDT, HDT++, OFR). An extensive experimental evaluation is done, in order to determine which approach works best, with somewhat mixed conclusions (probably use OFR, despite what the literature says, but then SPARQL support should be provided, if the latter is needed immediately, use k2).

Overall, the paper is nicely written and fills in a gap in the literature. I believe that some points could have been better explained (especially what is actually being done and measured), but I do believe that the paper is acceptable in its current shape.

I provide some specific comments and suggestions next:
- It should be made clearer in the introduction whether only the id to URI dictionary is compressed, the graph structure, or both. It becomes clear later, but stating it here would make things much clearer.
- It is not explained in the k2 tree construction whether the intermediate 0/1 matrix poses issues in terms of size.
- A bit of intuition on SPS would be nice. As it stands, I am wondering whether it is a bit of a crude metric.
- It would be useful to also measure memory footprint of different methods. I am particularly wondering if the k2 tree comment above makes the index creation memory heavy.
- Perhaps a discussion on why the results of [12] are different from the ones presented here might be useful. I do understand this would be speculation, but maybe looking into the data/setup would reveal something.
- It should be better explained how the decompression time is measured. Usually in this sort of compression we would be interested in retrieving/searching the compressed indices. Namely, in order to answer a SPARQL query, we would be interested in access times for a triple, finding all neighbors of a node with a certain label, retrieving an URI from an id and vice versa. These measures would shed more light on whether the proposed compression can work well withing a querying framework.


**Anonymity:**

Yes, I would like my review to remain anonymous.

**Reuse And Availability:**

4: High

**Strong Points:**

- The studied problem is relevant.
- Implementation of the studied methods is available.
- Decent experimental design (with some gaps; see below).

**Subreviewer:**

I submitted this review.

**Weak Points:**

- The decompression measure is a bit confusing, and the retrieval metrics (see the general comments above) are not studied in depth.
- It should be better explained what is actually done/compressed in the Introduction.

---

> ### Author Rebuttal · Authors · 2021-01-30
>
> We want to thank the reviewer for the detailed review.
>
> 1., 3. The suggestions are very welcome and we want to follow them when creating the final version of the paper.
> 2., 4. An intermediate sparse matrix is stored as a set of points, where the points (row, column) represent a cell with value 1. During implementation & evaluation this matrix representation did not create any memory issues. The memory intense part of the k2 algorithm is the creation of the k2 tree from these points but not the points themselves. Especially if several k2 trees were created in parallel the memory consumption was as expected way more.
> 5. From what we gathered, the k2 implementation we use in our experiments seems to achieve a better compression than the variant that Maneth et al. use in their experiments [12]. In addition, gRePair would typically not store the internal nodes but would use the order of node ids to derive which node has been removed at which time. However, this is not possible in our setup since we use a dictionary that defines the ids based on its internal structure. From our point of view, this is an unclear point since Maneth et al. claim that their approach would work with any dictionary. We contacted them to clarify this point but couldn’t get an answer until now. Hence, this second reason is just a speculation.
> 6. The decompression time is the time that is necessary to decompress the complete graph and bring it back into its original input format. Unfortunately, it is not possible for us to measure the time to access single triples since this is not implemented for OFR, gRePair, k2 and RDFRePair.

---

### Official Review · AnonReviewer2 · 2021-01-13
**Interesting negative results but requires technical clarifications and more explanation on *why* results are negative**

**Rating:** 1
**Confidence:** 4
**Impact:** 2
**Design And Technical Quality:** 4

**Review:**

# Summary

This paper explores compression techniques for RDF graphs. Specifically, the main proposal explores a variant of an existing method called gRePair, based on encoding frequent digrams (pairs of edges) in the graph separately in a compressed way, thereafter using k^2 trees for each edge label (predicate) to encode the structure of the graph. The authors call this proposal "RDFRePair". The paper defines the encoding used, the method of compression/serialisation, and the method of decompression used. Experiments are run on a variety of datasets, comparing RDFRePair (and its underlying method "gRePair") with existing compression methods: a plain k^2 encoding of the RDF graph, HDT, HDT++, and OFR. Results show that RDFRePair and gRePair do not scale well for larger graphs. Also the proposed approach is not competitive in terms of compression ratio nor compression times, being less efficient and less scalable than the baseline methods, which also have better compression ratios. The paper is thus submitted as a Negative Results Paper.

# Strengths

1. The proposed approach is based on an existing algorithm (gRePair) that has achieved good compression over graphs, and has even had encouraging results over RDF graphs.

2. The experiments consider a good diversity of datasets and baselines.

3. Aside from some technical details that I found unclear, the paper is well written.

# Weaknesses

1. Some technical details are vague and were difficult to follow for me. I can follow the general idea of the RDFRePair encoding, but I found the text insufficiently precise to understand the full details.

2. There are some issues in the experimental section that should be discussed. Also the call mentions that negative results papers should provide "A thorough analysis of why the results are negative", which I don't think is provided.

3. As a research paper (even considering the negative results), the novelty is quite unclear. The paper feels almost more suited to be published as an experimental paper as the new algorithm proposed seems to be a minor variation of existing techniques.

# Verdict

Regarding W3, it may not be a critical weakness, depending on the interpretation of the call. The call for negative results states that such papers could even report on textbook methods, from which I interpret that, in fact, the call is targeting more experimental papers rather than ones that satisfy typical novelty criteria for the research track. Still though, I think the paper would benefit from clarifying certain aspects of the novelty (discussed below in more detail).

Overall, I think that if these three weaknesses could addressed, the paper could be published. I think that weaknesses 1 and 3 mainly requires some technical clarifications, while regarding weakness 2, I expect that the authors can extend the discussion on why the results were negative. I think that all three weaknesses could be addressed for the camera-ready version. So I lean slightly towards an accept.

**Anonymity:**

No, I would like my review to be deanonymized.

**Reuse And Availability:**

5: Very High

**Strong Points:**

1. The proposed approach is based on an existing algorithm (gRePair) that has achieved good compression over graphs, and has even had encouraging preliminary results over RDF graphs. The negative results call specifically solicits papers on "Ideas that look unarguably plausible, but do not work out", and/or "Textbook methods (or methods suggested in highly cited papers) applied to real world problems with surprisingly negative outcomes". I think the paper satisfies both aspects.

2. The experiments are thorough, with a good mix of datasets and baselines. Though some additional baselines and datasets could be considered (the paper discusses the importance of querying, but that is not fully developed, and hence one might hope for reference values involving DEFLATE, bz2, etc.), overall I think the experimental results are perhaps the strongest part of the paper.

3. The paper is well-motivated, in general well-written (aside from some technical details and other minor issues listed below) and easy to follow. The referenced works appear reasonably complete; at least I am not aware of concrete missing references.

**Subreviewer:**

I submitted this review.

**Weak Points:**

1. Some technical details are vague and were genuinely difficult for me to follow: I think some technical clarifications are needed regarding RDFRePair. Providing an example would be welcome as the overall intuition is missing, which makes it even more problematic that the technical details are not precise. (In terms of space, the k^2 tree construction appears pretty standard and could be dropped or shortened perhaps to make room for an illustrative example.) Specific issues include:
    - The paper refers to digrams in two different ways. First in definition 4, a digram is defined as a pair of (directed, unlabelled) edges. There are then 33 different digram types with internal/external nodes. This makes sense, but then "digram occurrence is defined as the occurrence of such a digram in a given graph"; what does that actually mean? Does this now include the predicates/edge labels? Maybe it means occurrences of types of digrams? It also suggests that digrams are not pairs of edges here, but are types of digrams. In Definition 5, it then becomes unclear in which sense the term "digram" is used; do we have 33 elements of D, or do we have the infinite set of all possible sets of pairs of nodes? It becomes hard to follow what precisely the different references of "digram" refer to.
    - In Section 4.2, I did not understand step 5 very well. A digram I understand to consist of two edges, which is replaced with one edge labelled d. What are the nodes of this edge? [I have realised that it is presumably the external nodes, but this should be explicitly clarified and it is still not clear what the second node will be if there is only one external node.]
    - In Section 4.2, I am not sure why step 3 is required; if digrams are replaced in order of frequency, then overlapping digrams will no longer be present by the time the algorithm comes to replace them?
    - What is the start graph in Section 4.3? Is it the result of the algorithm of Section 4.2?
    - In Section 4.3, "before it is merged with all other paths of the matrix" What are the paths of the matrix? (I presume this refers to the paths in the k^2 tree.)
    - In Section 4.4, "The size flag uses two bytes" In the diagram, size is shown as 2 bits. I guess this should read two bits? (Why is one bit not sufficient if the idea is to switch between 1 byte and 2 byte IDs?)
    - In Section 4.4., I struggled to understand the digram serialisation. Per definition 4, a digram is a set of pair of nodes. But the serialised digram now contains edge label ids, so it does not really represent a digram, but something different? It also mentions that all occurrences of the digram are encoded, but it is not clear if this means since a digram is a set of pairs of nodes, and if we fix the edge labels, we end up with a single pair of labelled edges. I think what I understand from this is that each serialisation encodes a type of digram and pair of edge labels, listing all nodes that "instantiate" the type of digram with those edge labels. I still do not understand "The last part of the digram ... taking any additional space".

2. The experimental results have certain possible pitfalls that should be discussed, and also, overall, the paper fails to pinpoint why the results are negative, which the call highlights to be an important element for such papers:
    - It appears from the experimental results that gRePair and RDFRePair have trouble scaling to large graphs. This is a negative result as compression is most important for large graphs. I think it is fine to present negative results, but I would have liked more discussion of this. I presume the issue is RAM (noting from Footnote 1 that a machine with 1TB of RAM was used in one experiment), so I would encourage the authors to discuss this limitation in more detail in terms of what is most greedy with the RAM, and if there is a possibility to perhaps use secondary storage, for example.
     - The Wilcoxon significance tests are not described in detail, but depending on how they were conducted, the results might be a bit flawed. In particular, some of the datasets are split into segments, with some datasets having more/fewer segments. If the compression ratios for individual segments were considered for the significance tests, it would "favour" algorithms that tended to perform better for datasets with more segments. Discussing this issue would be important. (Also, why was the significance level changed in Table 3?)
    - Both gRePair and RDFRePair are outperformed in terms of compression rates and times by baseline methods, which is clear in the results, but it is not clear from the discussion *why* this might be the case. Per the call, it is important not only to have negative results, but also to highlight *why* the results are negative, and in particular, to highlight potentially flawed assumptions that led to better expectations for the results. This is largely missing from the paper at the moment.

3. The novelty is unclear. I think even if novelty is not a key requirement of a negative results paper, I still think it would be good to clarify some details. In particular, the paper mentions that gRePair has already been evaluated on RDF graphs, but the authors mentioning porting it to RDF graphs as a novel aspect of their work. Also gRePair is evaluated over RDF in the paper itself. Hence it is not clear how RDFRePair actually differs from gRePair, which is something I think that should be clarified early on in the paper. It is mentioned in the middle of the paper that the idea is to not consider cases with three external nodes; is that the only difference?

---

# Minor comments:
- General: the paper would be more readable if you could avoid using numeric references as nouns; for example: "[16] proposes the usage" reads as "sixteen proposes the usage", whereas a much more readable phrase would be "Wang et al. [16] propose the usage". Also instead of phrases like "The authors of [13]", better to write "Pan et al. [13]", which is equally as readable but more informative. This continues throughout the paper. Another example is "combines it with the dictionary of [7]"; it would make the reader's life easier to simply write "combines it with the dictionary of HDT [7]". In general, I would suggest to avoid relying on numeric references as nouns (rather consider them as parenthetical, which they are supposed to be) and to identify individual works other than numerically; this way the reader does not waste time unnecessarily hunting for references to understand the text.
- "This general concept is wide[] spread"
- "searches for edge digrams" A brief description of edge digrams would be welcome here, otherwise it is difficult to understand the idea of the approach.
- "are called quadrant[s]"
- " as potential digram[s]"
- "to which exten[t]"
- "watdive" -> "WatDiv"
- "and [in] part RDFRePair were not able"

---

> ### Author Rebuttal · Authors · 2021-01-30
>
> We thank the reviewer for the detailed review.
>
> Weak Point 1.:
> While a digram type is the abstract digram independent of a graph, the digram is a digram type including the edge labels. So there are 33 digram types but many more digrams. A digram occurrence is the instance of such a digram in a specific graph.
> In Definition 5, D is the set of all digrams, i.e., the set of the combinations of a digram type and 2 edge labels that occur in the given graph.
> We will emphasize these differences and check that the correct wording is used throughout the paper.
> The external nodes of a digram are the nodes that are connected by the new edge. We will clarify that.
> For our current implementation, step 3 is not really necessary and we will remove its description to get more space for other pieces of information.
> We implemented the step since it allows the usage of other algorithms to retrieve the best set of non-overlapping digrams, which may result in a slightly better compression (Note that this is the travelling salesman problem though). However, the presence of this step had no significant negative effect on the runtime during our experiments.
> Yes it is the graph G which is the result of algorithm 4.2. We mention it explicitly.
> Each of these individual paths represents one path root to leaf in the k^2 tree. We will clarify that.
> Yes it should be two bits, thanks for pointing that out. One byte is not sufficient as we allow 1 byte, 2 byte and 4 byte IDs.
> A digram is a pair of two edges per Def. 4. Hence we save the two edge label IDs. As we have multiple occurence of this digram that we save the internal nodes which we also removed from the graph. To decompress correctly we have to assure that we can map the internal nodes to the correct occurence in the graph (otherwise the decompression wouldn’t yield the original graph). To assure that we can map the internal nodes correctly we save them in the mentioned order, thus in decompression we know implicitly which internal nodes belong to which digram occurrence.
> Weak Point 2.:
> For gRePair and RDFRePair, the runtime was the main issue. From our experiments, we derived that subsets of 50k triples are sizes that can be handled by the grammar-based algorithms in a reasonable amount of time. We will emphasize this in the discussion.
> Yes, more splits could automatically lead to a higher influence on the result. This is an important point. We will mention it in the discussion.
> From what we gathered, the k2 implementation we use in our experiments seems to achieve a better compression than the variant that Maneth et al. use in their experiments [12]. In addition, gRePair would typically not store the internal nodes but would use the order of node ids to derive which node has been removed at which time. However, this is not possible in our setup since we use a dictionary that defines the ids based on its internal structure. From our point of view, this is an unclear point since Maneth et al. claim that their approach would work with any dictionary. We contacted them to clarify this point but couldn’t get an answer until now. Hence, this second reason is just a speculation.
> Weak Point 3.:
> First, Maneth et al. [12] evaluate their approach on RDF graphs and compare it with k2 trees. However, their prototypical implementation comes with some drawbacks. First, the decompression does not seem to be reliable. Second, it does not use any dictionary but assumes that node and edge labels have been transformed into ids. Our implementation ensures that the implementation covers the complete functionality of an RDF compression algorithm and that the compression fits to the dictionary. Apart from that, RDFRePair has some differences in its implementation. RDFRePair does not make use of hyperedges. The implementation of the search for digrams is slightly different. In gRePair the search for digrams begins from one vertex and can get different sets of digrams dependent on where the search started. In contrast, RDFRePair traverses over all vertices and retrieves every digram. In addition, the digrams are stored in a different way that tries to reduce their size further.

---

> > ### Comment · AnonReviewer2 · 2021-01-31
> > **Response to rebuttal [maintaining score]**
> >
> > I appreciate the clarifications of the authors. As a meta-comment, I found the rebuttal a bit difficult to parse as it responds to various points in one "wall of text" without delimiters for sub-points. Perhaps there are length limits, but still I would encourage the authors to spend some of that limit on formatting in future.
> >
> > The clarifications for Weak Point 1 seem fine to me. I think this mainly requires clarification in the paper and more careful use of terminology (in particular, surrounding "digram")
> >
> > Regarding Weak Point 2: "gRePair and RDFRePair, the runtime was the main issue". Understood (it's not a RAM issue). The clarification on scale limits are also appreciated. But overall it would be good to try to get to an underlying reason for the long runtimes, or to indicate what parts of the process form the "choke points". Also I did not fully understand the explanations regarding compression rates, but hopefully the authors can clarify this in the paper.
> >
> > As for Weak Point 3, I think this requires some clarification in the paper.
> >
> > In light of the rebuttal, I maintain my original score of Weak Accept.

---

### Official Review · AnonReviewer5 · 2021-01-14
**Nice initial work towards more efficient grammar-based compression on RDF**

**Rating:** 1
**Confidence:** 5
**Impact:** 3
**Design And Technical Quality:** 3

**Review:**

The paper addresses the topic of RDF compression, which has gained increasingly attention, mostly since the use of HDT and approaches like TPF using it. In particular, authors focus on applying grammar-based compression. To do so, they present RDFRePair, an adaptation of gRePair (a graph grammar-based compression) and an implementation of K2 trees. Their evaluation shows that this latter shows the best ratios w.r.t. the state of the art, while gRePair is well positioned.

The paper is overall interesting as RDF compression has still a long way ahead. I particularly appreciate the evaluation where statistical measurements are taking into account for comparison.

I have, however, some doubts that hopefully can be resolved during the rebuttal:

- The contribution of the adaptation of gRePair is unclear. Besides the addition of the dictionary (which reuses [7] and should be straightforward) and implementation tricks with hyperedges, could authors clearly state the main contribution?

- I have the same concerns with K2 trees. How is the author's contribution different than the existing ones [1] ? I will also suggest authors to have a look at the missing reference which updates [1]. It also proposes additional indexes for querying:
  * Sandra Álvarez, Nieves R. Brisaboa, Javier D. Fernández, Miguel A. Martínez-Prieto, Gonzalo Navarro. Compressed vertical partitioning for efficient RDF management. Knowledge and Information Systems, 44(2), 439-474, 2015. ISSN 0219-1377

- The claim "no previous work has addressed the concrete task of porting and comparing the current state of the art in graph compression with the current reigning RDF compression algorithms." is incorrect. While it might be true for grammar-based compression, it is absolutely false for graph compression. The work in [1] is applying k2trees to RDF, and K2trees was designed to compress web graphs (the graph of links between pages).

- The Dictionaries of HDT [7] are optimize for consumption, not space. Please have a look at the following paper, where different tradeoffs are shown:

* Martínez-Prieto, M. A., Brisaboa, N., Cánovas, R., Claude, F., & Navarro, G. (2016). Practical compressed string dictionaries. Information Systems, 56, 73-108.

- The section 4.6 regarding querying is a bit disappointing. I would recommend authors to downgrade the expectations since the introduction as authors claim that "RDF compression algorithms achieve better compression [...] in a manner which still allows for querying". IMHO RDF compression can be only compression and not querying, but authors should first fix this claim and in any case warn about future and eventual querying capabilities. This would be fair for other approaches, such as HDT that sacrifices compression for query performance.



**Anonymity:**

Yes, I would like my review to remain anonymous.

**Reuse And Availability:**

4: High

**Strong Points:**

- New adaptations of Grammar-based compression to RDF
- Available code
- Statistical comparison in the evaluation

**Subreviewer:**

I submitted this review.

**Weak Points:**

- Unclarity in the adaptations of gRePair and K2 trees
- Expectations regarding querying
- Evaluation with relatively small datasets

---

> ### Author Rebuttal · Authors · 2021-01-30
>
> We thank the reviewer for the detailed review.
>
> Our contribution is to provide a comparable implementation of the algorithm that can be used for repeatable evaluation. As stated in the paper, the prototypical implementation came with several drawbacks. For example, the decompression of our implementation is reliable and allows us to check for the correct compression and decompression of the data.
> First, thank you for the reference. We will add it to the final version of the paper.
>
> We know that k2-trees have been applied to RDF graphs in the past. However, we were surprised by our results that show better compression results for k2-trees than for most of the other approaches, which contradicts the results of other papers (e.g., Maneth et al. [12] or Hernandez-Illera et al. [9]). Unfortunately, we couldn’t get any open source implementation of a k2-trees-based RDF compressor. So we are not sure to which extent our implementation differs from the implementations that have been used by the related work.
>
> We made this statement with respect to the current state of the art which is (to the best of our knowledge) gRePair. Thanks to your explanation, we understand that the statement might be misleading. We will adapt it to avoid this kind of misunderstanding.
> Thank you for this reference.
>
> We are aware of this tradeoff and agree that it should be mentioned in the paper. Our main problem is that implementations to query data that is compressed with ORF or gRePair is theoretically possible (as explained in their publications) but not implemented. Hence, a comparison of these approaches with respect to query runtimes is not possible within our work as we would have to implement all the missing query engines.

---

> > ### Comment · AnonReviewer5 · 2021-02-01
> > **Thanks for the answers**
> >
> > I would like to acknowledge and thank the authors for the answers in the rebuttal.
> >
> > I hope authors can consider all these points in a revised version

---

### Official Review · AnonReviewer4 · 2021-01-14
**Important topic, but research contributions and evaluation need improvement**

**Confidence:** 5
**Impact:** 2
**Design And Technical Quality:** 2

**Review:**

This paper presents the following three contributions:

1. RDFRePair is the main contribution of this paper. It is an implementation of the gRePair compression algorithm introduced in 2018 by [1]. The goal of this contribution is to improve the implementation of this grammar-based compression algorithm, in order to optimise it for RDF graphs. While this seems like a promising idea for advancing the state-of-the-art of RDF graph compression techniques, the reader discovers at Section 5.2 that both the compression algorithm (gRePair) and its RDF implementation introduced by the authors (RDFRePair) do not scale at all. Looking at Figure 4, we can see that this algorithm could not be applied on 12 out of the 40 evaluated datasets, including the "geo_coordinates_en" dataset containing 2.3 million triples. My concern: what is the benefit of implementing an existing graph compression algorithm, and introducing it to the Semantic Web community, if it cannot compress RDF datasets larger than 2 million triples (within 2 hours)? For context, DBpedia and Wikidata (probably the most used RDF datasets) are three orders of magnitude larger than "geo_coordinates_en".

2. An implementation of another existing compression algorithm called k^2 introduced in 2009 by [2], for the goal of applying it on RDF graphs.

3. A large-scale evaluation comparing gRePair, RDFRePair and k^2 with three RDF based compression algorithms: HDT, HDT++, OFR. Since this is a negative results paper, the experiments show that the grammar-based compression algorithm described in this paper is outperformed by existing RDF graph compression algorithms such as OFR, w.r.t. both compression and decompression time, and compression ratio. The authors could not identify any significant correlation between the performance of algorithms and features of the considered datasets. My concern with the conducted evaluation is not the negative results. Instead, my concern is regarding the evaluation metrics that the authors use to show which compression algorithm is more efficient for RDF graphs, which are compression/decompression time and compression ratio. While these metrics are interesting to compare between the different algorithms, I think that there are far more important ones that make the difference between using one compression algorithm over the other, that were not considered by the authors. As a frequent user of HDT, I will rely on the HDT website and paper [3,4] to list some of these metrics: (i) scalability, (ii) memory usage, (iii) query performance, (iv) availability of libraries. I don't think that one can compare RDF graph compression algorithms and draw any conclusions on their performance without considering some of these metrics. As an RDF user, what is the benefit to me if the compression ratio of an algorithm X is is slightly better than algorithm Y, but X can't scale to large datasets, X does not provide me with tools to query the dataset or has worse querying performance, and if X requires significantly more memory to generate the index and query the dataset. After all, I believe the main cost on the user in compressing and querying large RDF graphs is memory related, and in my opinion much less related to storage and compression speed.

Conclusion: The authors are trying to shed light on an interesting and important aspect of manipulating RDF graphs. They are encouraged to pursue this work further and improve their evaluation by considering additional metrics (such as the ones listed above), in order to fairly compare the existing graph compression algorithms, and advance the state-of-the-art.


[1] Maneth, Sebastian, and Fabian Peternek. "Grammar-based graph compression." Information Systems 76 (2018): 19-45.

[2] Brisaboa, Nieves R., Susana Ladra, and Gonzalo Navarro. "k 2-trees for compact web graph representation." International symposium on string processing and information retrieval. Springer, Berlin, Heidelberg, 2009.

[3] Fernández, Javier D., et al. "Binary RDF representation for publication and exchange (HDT)." Journal of Web Semantics 19 (2013): 22-41.

[4] https://www.rdfhdt.org/technical-specification/#numbers

___

**AFTER REBUTTAL**

In my review, I pointed out two major problems in the paper as the reason for my rating:

(P1). "What is the benefit of implementing an existing graph compression algorithm, and introducing it to the Semantic Web community, if it cannot compress RDF datasets larger than 2 million triples."

(P2). "My concern is regarding the evaluation metrics that the authors use to show which compression algorithm is more efficient for RDF graphs, which are compression/decompression time and compression ratio. While these metrics are interesting to compare between the different algorithms, I think that there are far more important ones that make the difference between using one compression algorithm over the other, that were not considered by the authors. As a frequent user of HDT, I will rely on the HDT website and paper [3,4] to list some of these metrics: (i) scalability, (ii) memory usage, (iii) query performance, (iv) availability of libraries. I don't think that one can compare RDF graph compression algorithms and draw any conclusions on their performance without considering some of these metrics.

In the rebuttal, the authors start with the following sentence:

> The question with respect to the benefit of introducing a grammar-based compression algorithm into the Semantic Web community is exactly the question the paper tries to answer.

But the authors are answering a different question than (P1). I still don't understand the motivation of the authors in implementing the grammar-based compression algorithm gRePair, when the authors know that this approach does not scale, based on their following statement in Section 5.2 of the paper: "*Note that gRePair and to a part RDFRePair were not able to handle large datasets in our experiments (Section 5.2)*". I assume that before starting their paper, the authors have tested the initial algorithm that they are basing their paper on, before implementing it for RDF graphs and comparing it to other RDF compressors. So the authors knew that the initial algorithm gRePair does not scale, but decided to implement it for RDF graphs anyway? Please note that while scalability might be a minor detail for most papers/approaches, it is an essential feature for compression algorithms, and the main motivation for having a compression algorithm in the first place. Why would I need to compress datasets smaller than 2 million triples? A 2 million-triple dataset is less than 400MB of size. I have to mention again that commonly used RDF datasets (e.g. DBpedia and Wikidata) are three orders of magnitude larger than this, and they are being compressed and queried using HDT for more than 10 years (https://www.rdfhdt.org/datasets/). How far is the introduced approach in catching-up with a 10 year-old technology for compressing and querying commonly used RDF datasets? Given these facts, the answer should have been clear even before the start of the paper, with a simple test of gRePair on any dataset larger than 2 million triples.

The authors continue their rebuttal by stating the two main benefits a graph compression could offer:

> From our point of view, there are mainly 2 benefits a graph compression algorithm could offer. First, it could answer queries faster than others. This is already discussed in [1] from a theoretical perspective. Maneth et al. conclude that a grammar-based compression would answer most queries most probably slower (apart from path queries which might be faster).

So the first benefit of grammar-based compression algorithm is their query performance, based on a discussion in [1]. Firstly, how is it possible that the main benefit of the introduced approach is not even mentioned in the paper? Most importantly why didn't the authors evaluate this claim? I thought that was the point of the paper, to introduce this grammar-based algorithm to the Semantic Web community, and show its benefits/drawbacks by comparing it to state-of-the-art approaches that work for RDF graphs. That's my point in (P2), that authors need to add these measures in the paper so readers can actually see whether this approach is indeed better than the existing ones. Otherwise, what is the contribution of the paper? Just comparing the compression ratio of some approaches?

> Second, the compression ratio could be better. The results presented in [1] seem to suggest it. However, although RDFRePair has a better performance than the gRePair implementation, our work shows that other existing compression algorithms lead to a better compression ratio.

Therefore, the second benefit of grammar-based compression algorithm is not actually a benefit, hence the reason why this is a negative results paper. The authors continue with:

>  Hence, we can already state that further optimizations with respect to the runtime or the used memory won’t lead to any significant impact for the Semantic Web community.

If I understood correctly, the authors are saying that the approach they're testing in this paper has already performed so bad in terms of scalability and compression ratio, that there's no point to test these additional important metrics anymore.

>  The tools support is very important in practice. However, since this is a research paper that looks at new ground (i.e., applying grammar-based compressions to RDF graphs), taking into account the existing support does not seem to be very relevant to answer our research questions.

Fair point.


**Conclusion:**

In the rebuttal, the authors claim that grammar-based approaches are interesting to the Semantic Web community, because according to [1], they have two main benefits:

(a) Better compression ratio, but this turns out to be not true, hence the negative results.

(b) Better query time, which the authors do not mention in the paper as a benefit, and do not even consider testing whether this claim might turn out to be (also) not true. In addition, how can the authors say in the rebuttal "could answer queries faster than others", if you don't compare it with these other approaches. What I mean is that neither in [1], neither in this paper, there is an experiment comparing query time performance of gRePair with the other RDF compressors. So how can the authors claim that it can answer queries faster? It might be fast, but you can't claim it is faster if you don't compare.

In addition, both the original (gRePair) and the implemented (RDFRePair) grammar-based approach do not scale to datasets larger than 2 million triples. Therefore, it is still not clear to me what is the benefit of introducing a compression algorithm to the Semantic Web community and presenting these negative results regarding only the compression ratio, knowing that it can't handle datasets three order of magnitude smaller than what RDF compressors have been doing for the last 10 years.

For the above reasons, my rating remains the same, as I believe this paper falls in the category of "What a Negative Results paper is not", according to https://2021.eswc-conferences.org/call-for-papers-research-track/




**Anonymity:**

Yes, I would like my review to remain anonymous.

**Rating:**

-2: Reject

**Reuse And Availability:**

3: Medium

**Subreviewer:**

I submitted this review.

---

> ### Author Rebuttal · Authors · 2021-01-30
>
> We thank the reviewer for the detailed review.
>
> The question with respect to the benefit of introducing a grammar-based compression algorithm into the Semantic Web community is exactly the question the paper tries to answer. From our point of view, there are mainly 2 benefits a graph compression algorithm could offer. First, it could answer queries faster than others. This is already discussed in [1] from a theoretical perspective. Maneth et al. conclude that a grammar-based compression would answer most queries most probably slower (apart from path queries which might be faster). Second, the compression ratio could be better. The results presented in [1] seem to suggest it. However, although RDFRePair has a better performance than the gRePair implementation, our work shows that other existing compression algorithms lead to a better compression ratio. Hence, we can already state that further optimizations with respect to the runtime or the used memory won’t lead to any significant impact for the Semantic Web community.
> The same is reflected in the metrics. We measured the compression ratio and scalability and we could show that the grammar-based compression fails in them. Measuring query speed does not seem to be fruitful because of the discussion in [1]. The tools support is very important in practice. However, since this is a research paper that looks at new ground (i.e., applying grammar-based compressions to RDF graphs), taking into account the existing support does not seem to be very relevant to answer our research questions.

---

### Decision · Program_Chairs · 2021-02-23

**Decision:**

Accept

**Comment:**

This paper reports partially negative results on grammar based compression of RDF graphs. While the presented RDFrepair algorithm, a port from an existing grammar-based graph compression algorithm works better than the original graph algorithm on RDf graphs, it is outperformed by other algorithms such as k^2 trees in compression size, but not in timewise performance  for compression and decompression, where again other another algorithm OFR performs better.
While it was disputed, whether this is actually a negative result (one reviewer particularly agrues the authors could have made more attempts to understand why their grammar-based method adapedted to RDF did not perform better and their might be still additional potential) , the thorough comparison and analysis of existing approaches along these different dimensions was appreciated by the reviewers as such, plus this potential deficiency seen as potential for future works in the otherwise not yet very much explored space of such grammar-based compression algorithms for RDF. As such, we propose to accept this paper as a controversial, but hopefully inspiring contribution for more work in this direction.

We recommend the authors to, as some reviewers suggest be more clear about limitations of the current approach, e.g. to downgrade the expectations for querying or resp. be more clear about compression vs. query performance of different approaches and add clarifications that have been asked for in the reviews, in particular extend the discussion on why the results were negative.